# Robust SARS-CoV-2 Antibody Responses in Asian COVID-Naïve Subjects 180 Days after Two Doses of BNT162b2 mRNA COVID-19 Vaccine

**DOI:** 10.3390/vaccines9111241

**Published:** 2021-10-25

**Authors:** Chin-Shern Lau, Soon Kieng Phua, Ya-Li Liang, Helen May-Lin Oh, Tar-Choon Aw

**Affiliations:** 1Department of Laboratory Medicine, Changi General Hospital, Singapore 529889, Singapore; michael.lau@mohh.com.sg (C.-S.L.); soon_kieng_phua@cgh.com.sg (S.K.P.); yali_liang@cgh.com.sg (Y.-L.L.); 2Department of Infectious Diseases, Changi General Hospital, Singapore 529889, Singapore; helen.oh.m.l@singhealth.com.sg; 3Department of Medicine, National University of Singapore, Singapore 119077, Singapore; 4Academic Pathology Program, Duke-NUS Medical School, Singapore 169857, Singapore

**Keywords:** SARS-CoV-2, mRNA vaccines, spike antibodies, neutralizing antibodies

## Abstract

Background: Subjects with previous COVID-19 have augmented post-vaccination responses. However, the antibody response in COVID-naïve subjects from Southeast Asia is not well known. Methods: 77 COVID-naïve vaccinees were tested with a full antibody panel [spike antibodies (total (T-Ab), IgG, IgM) and neutralizing antibodies (N-Ab)] pre-vaccination, 10 days after dose 1, and 20/40/60/90/120/150/180 days after dose 2. Results: 10 days after dose 1, 67.6% (48/71)/69.0% (49/71) were T-Ab/IgG positive; only 15.5% (11/71)/14.1% (10/71) were N-Ab/IgM positive. While all (100%) subjects had brisk T-Ab, IgG and N-Ab antibody responses 20 days after complete vaccination, only 79.1% (53/67) were IgM positive. At 180 days (*n* = 8), T-Ab/IgG/N-Ab were still reactive (lowest T-Ab 186 U/mL, IgG 617 AU/mL, N-Ab 0.39 µg/mL), but IgM was negative in all samples. Spike antibody thresholds of T-Ab 74.1 U/mL (r = 0.95) and IgG 916 AU/mL (r = 0.95) corresponded to N-Ab reactivity (>0.3 µg/mL). Non-linear regression analysis showed that N-Ab would decrease to 0.3 µg/mL by 241 days, whereas T-Ab/IgG would need 470/163 days to reach titers of T-Ab/IgG associated with a N-Ab 0.3 µg/mL (76.4 U/mL and 916 AU/mL respectively). Conclusions: The antibody responses of T-Ab, IgG and N-Ab remain high and durable even at 180 days. N-Ab titers are expected to remain reactive up to 241 days post-vaccination.

## 1. Introduction

Severe acute respiratory syndrome coronavirus type 2 (SARS-CoV-2) antibody levels decline several months after coronavirus disease 2019 (COVID-19) infection/vaccination. However, some studies report that the increase in nucleocapsid antibodies (Nuc-Ab) can last up to 32 weeks post-infection [1], with detection rates lasting at least 7.5 months. Moreover, antibody kinetics may also vary according to the assay used. In one study [2], six different analytical kits (Roche total Nuc-Ab + spike antibodies (S-Ab), Liaison IgG S-Ab, Vitros total + IgG S-Ab, and Phadia IgG S-Ab) were used to assess antibody responses up to 10 months after COVID-19 infection. All assays except the Phadia IgG S-Ab demonstrated >90% positivity rates even at 300 days after COVID-19 infection. Although further research is required to decipher the protein coded by the mRNA vaccines, the antibody responses to the vaccine are well documented. In our previous preliminary study [3], 90 days (3 months) after two doses BNT162b2 SARS-CoV-2 mRNA vaccine, COVID-naïve subjects only experienced a small decrease in total, IgG S-Ab and neutralizing antibodies (N-Ab), with all titers still reactive at the end of three months. In another Belgian multicenter study [4] of 200 healthcare professionals who received two doses of BNT162b2 vaccine, S-Ab levels also remained high at 90 days post-vaccination, even in initially seronegative subjects (mean 1262 U/mL). Furthermore, IgM levels post-vaccination remains unclear, as most studies have not studied IgM responses.

N-Abs are a subset of the humoral response to a viral infection. The N-Abs elicited by SARS-CoV-2 infection/vaccination are specific to the receptor binding domain of the viral spike protein [5]. Antibodies directed against the S1 receptor binding domain account for around 90% of serum neutralizing activity, with higher levels associated with decreased disease severity [6], with many serology assays positively correlating with SARS-CoV-2 neutralization activity [7]. Serological assays may be a useful marker in the assessment of protection against COVID-19, although the neutralizing activity of N-Abs has thus far only been shown in vitro. One study [8] showed that N-Ab activity remained high 180 days (6 months) after a second dose of mRNA-1273 vaccine in 33 healthy participants. However, N-Ab titers were assessed using their own enzyme-linked immunoassay (ELISA) method, as part of an ongoing phase one trial. Indeed, many studies involving N-Ab assessment utilize ELISAs, virus neutralization test kits, or viral culture methods, which may require specialized equipment or facilities for the handling of live viruses. N-Ab chemiluminescent immunoassay (CLIAs) is now available on a fully automated platforms, which are safer and easier to operate. However, there is a paucity of studies that utilize these assays to assess the long-term trend of N-Ab post-vaccination.

In addition, there is a paucity of literature exploring the antibody response to vaccination with the BNT162b2 vaccine more than 3 months after two doses of vaccine in Southeast Asia. We thus examined the titers of total, IgG, IgM S-Ab and N-Ab in COVID-naïve individuals after two doses of BNT162b2 vaccine up to 180 days, using automated immunoassay platforms.

## 2. Methods

### 2.1. Participants

Between January to September 2021, we recruited 77 healthcare workers (HCWs) with no prior history of COVID-19 infection and tested their antibody levels at baseline (pre-vaccination), 10 days after the first dose of BNT162b2 vaccine, 20–40 days after the first vaccine dose (just prior to receiving the second dose), and 20/40/60/90/120/150/180 days after the second vaccine dose. Due to different vaccination schedules, the number of samples at each time point was different. The population age ranged from 24–70 (mean 40.0 ± 11.9 years), and 24.7% were male (19/77) with 75.3% females (58/77).

### 2.2. Methods and Materials

Serum at each time point was obtained and stored at −70 degrees Celsius if not immediately analyzed. Frozen samples were thawed for 1 h at room temperature just prior to analysis. Thawed samples were vortexed before analysis.

Both the Roche total S-Ab and Abbott IgG S-Ab can be converted to WHO international units based on user circulars provided by the manufacturers. For the Roche total spike antibody (T-Ab) BAU/mL = 0.97 × Roche value in U/mL, and for the Abbott IgG S-Ab BAU/mL = 0.142 × Abbott value in AU/mL. The Roche Elecsys Anti-SARS-CoV-2 S quantitative double-antigen sandwich electro-chemiluminescent immunoassay (ECLIA) (run on the Roche Elecsys e801 auto-analyzer) has a positive threshold of ≥0.78 BAU/mL (assay upper limit 243 BAU/mL, dilution range up to 1:100, limit of detection 0.31 BAU/mL, reported precision of 2.9% and 1.4% at 0.47 and 178 BAU/mL; reported assay sensitivity of 98.8% and specificity of 99.98%). The Abbott quantitative IgG S-Ab assay run on the Abbott Architect has a measuring range of 3.0–5680 BAU/mL, with ≥7.1 BAU/mL considered positive (reported precision 4.9% and 5.1% at 6.8 and 5115 BAU/mL; limit of detection 1.0 BAU/mL, reported sensitivity 66–99% and specificity 99.6%). IgM S-Ab was assessed on the Abbott Architect qualitative SARS-CoV-2 IgM S-Ab assay (positive cut-off index (COI) ≥ 1.0) whose performance was previously reported [9]. N-Ab was assessed with the Snibe competitive quantitative N-Ab assay run on the Snibe Maglumi, where sample N-Ab competes with angiotensin-converting enzyme 2 antigen immobilized on a solid phase for binding labelled recombinant SARS-CoV-2 S Receptor Binding Domain antigen to produce a light signal that is inversely proportional to the sample N-Ab. It has a measuring range of 0.05–30 µg/mL, with ≥0.3 µg/mL regarded as positive. The claimed interassay precision is 1.27% and 1.01% at 0.079 and 21.192 µg/mL, limit of detection 0.045 µg/mL, sensitivity of 100% and specificity 100%.

To ensure that subjects had no prior COVID-19 or asymptomatic COVID-19 infections during the study period, total and IgG Nuc-Abs were tested at each time point on previously evaluated SARS-CoV-2 Nuc-Ab assays [10,11]. In addition, all participants tested negative in our institutional RT-PCR screening exercises in May and June 2021. Our SARS-CoV-2 RT-PCR testing was performed using the Cobas SARS-CoV-2 qualitative assay on the Cobas 6800 System.

### 2.3. Statistical Analysis

Data were presented in either mean ± standard deviation or median (inter-quartile range) where appropriate. No indeterminate or missing results were used. We utilized Mann-Whitney U testing to compare the antibody titers between different population groups, with *p* < 0.05 considered statistically significant. Standard regression analysis was also performed to assess the agreement between S-Ab and N-Ab titers. We used MedCalc Statistical Software (version 20.008, MedCalc Software Ltd., Ostend, Belgium) for statistical analyses. To assess the half-life of the various antibodies post-vaccination, we utilized a simple non-linear regression model that correlated the log_10_ antibody levels to days post-vaccination (GraphPad Prism, version 9.2.0, GraphPad Software, San Diego, CA, USA). Predicted values from this model were then applied to the SydPath Half-Life Calculator [12] to determine antibody half-life.

Our institution’s IRB deemed this work exempt as this was part of seroprevalence survey and method evaluation. However, informed consent was obtained from all subjects involved in the study, as they needed to provide blood samples on multiple occasions. The study was conducted in accordance with the World Medical Association Declaration of Helsinki [13], and compliance with STARD guidelines is enclosed (see Appendix A).

## 3. Results

### 3.1. COVID-Naïve Status of All Subjects

At all time points, all subjects were negative for total and IgG Nuc-Ab. Furthermore, all subjects were also SARS-CoV-2 RT-PCR negative in May and June 2021. No subjects had any hospital admissions or experienced any COVID-19 related symptoms during the entire study period.

### 3.2. S-Ab and N-Ab Responses

Pre-vaccination, all subjects were spike antibody (S-Ab) (total, IgG and IgM) negative. 10 days after the first vaccination, the median S-Abs (T-Ab and IgG) were above reactive cut-offs, with 67.6% (48/71) T-Ab positive and 69.0% (49/71) IgG positive. However, only 15.5% (11/71) were N-Ab and 14.1% (10/71) IgM positive. 8 subjects were able to provide samples 20–40 days after their first dose of vaccination (just prior to receiving the second vaccine dose) and antibodies had significantly higher medians compared to day 10 post-first dose (T-Ab median difference 81 BAU/mL, *p* = 0.0008; IgG median difference 99 BAU/mL, *p* = 0.001; N-Ab median difference 0.27 µg/mL, *p* = 0.001) except for IgM (median difference COI 0.22, *p* = 0.16) (see Figure 1 and Figure 2, Appendix A).

20 days after the second vaccination, all antibodies except IgM (IgM 79.1% (53/67) positive) were positive in 100% of the subjects. All antibody titers at this point were significantly higher than 20–40 days after the first vaccine dose (T-Ab difference 2083 BAU/mL, 95% confidence interval (CI) 1281–2852, *p* < 0.0001; IgG difference 2067 BAU/mL, 95% CI 1274–2902, *p* < 0.0001; IgM difference COI 1.09, 95% CI 0.05–2.81, *p* = 0.04; N-Ab difference 2.84 µg/mL, 95% CI 1.92–4.63, *p* < 0.0001). Antibodies experienced a slow decline from that point onwards (see Figure 1). However, even at 180 days (*n* = 8), T-Ab/IgG/N-Ab were still elevated (lowest T-Ab 186 U/mL, IgG 617 AU/mL, N-Ab 0.39 µg/mL), while all IgM titers were negative (COI 0.03-0.83, median COI 0.14). The 180-day T-Ab/N-Ab titers were significantly higher than that at 20–40 days post-first dose (T-Ab median difference 638 BAU/mL, 95% CI 228–1429, *p* = 0.005; N-Ab median difference 0.62 µg/mL, 95% CI 0.07–1.23, *p* = 0.03).

The median IgG at 180 days post-vaccination was higher than that at 20–40 days after the first vaccine dose (263 vs. 116 BAU/mL). However, this did not achieve statistical significance (median difference 89 BAU/mL, 95% CI -344 to 290, *p* = 0.40).

### 3.3. Regression Analysis

Regression analysis showed a good correlation between T-Ab/IgG and N-Ab (T-Ab r = 0.95, IgG r = 0.95) (see Figure 3a,b). The agreement was less between IgM and N-Ab (r = 0.51). When the T-Ab and IgG are both expressed in WHO international units, the antibody values were numerically closer (WHO units: r = 0.93, Log(IgG) = 0.671 Log(T-Ab) + 0.886; manufacturer units: Log(IgG) = 0.671 Log(T-Ab) + 1.725).

Based on the equations derived from the linear regression analysis, at the reactive cut-off values for N-Ab of 0.3 µg/mL, T-Ab would correspond to 74.1 BAU/mL (or 76.4 U/mL in manufacturer units) and IgG would be 130 BAU/mL (or 916 AU/mL in manufacturer units) (see Figure 3a,b).

### 3.4. Estimation of Antibody Half-Lives

We created non-linear regression models of the antibody titers with time (see Figure 4), with T-Ab and IgG expressed in their normal, manufacturer recommended units (U/mL and AU/mL, respectively). Based on the equations of the non-linear regression models, the average half life for T-Ab/IgG/IgM/N-Ab were 90/33/28/56 days respectively (see Table 1). Based on our non-linear regression equation, the N-Ab would decrease to 0.3 µg/mL (reactive cut-off) at 241 days and the T-Ab/IgG would decrease to the corresponding levels (76.4 U/mL and 916 AU/mL, *vide supra*) by day 470/163.

### 3.5. Age and Gender Group Analysis

We also compared the antibody responses between sub-groups of age and gender at >90 days after complete vaccination (<50 years *n* = 18, ≥50 years *n* = 16; males *n* = 9, females n = 25). We did not find any significant differences in antibody levels between age or gender groups in all results >90 days (see Appendix A).

## 4. Discussion

We observed that all antibody responses slowly declined with time. Even though we used an Asian population, our antibody trend was similar to other studies that observed declining antibody titers in Israeli [14]/Caucasian [15] populations, with final antibody levels at 180 days still higher than reactive cut-offs. Compared to the study of a Belgian population [15] which used the same assays as we did, our final median T-Ab and IgG S-Ab levels were only slightly lower than theirs (T-Ab: 968 BAU/mL vs. our 897 BAU/mL; IgG S-Ab: 277 BAU/mL vs. our 259 BAU/mL), confirming that the pattern of the serologic response between populations is similar. There are concerns that this decline would translate into sub-optimal levels of protection, and no definitive antibody titers have been confirmed as protective. One study [16] estimated that a vaccine with an initial efficacy of 95% would only have an efficacy of 77% by 250 days, although it still would provide a considerable level of protection against severe COVID-19 at that point. Although we are unable to estimate the efficacy of the antibody responses in our population, all results for S-Ab (T-Ab and IgG) and N-Ab were still reactive despite the decline in antibodies even at 180 days post-vaccination. In addition, all our participants were COVID-naïve, and we rigorously confirmed the absence of even asymptomatic COVID-19 with negative Nuc-Abs and SARS-CoV-2 RT-PCR tests throughout the entire study period. Despite being seronegative, all T-Ab/IgG/N-Ab values at 180 days were still reactive, with the lowest T-Ab (180 BAU/mL) still 231× higher than the manufacturer recommended cut-off. Even the lowest N-Ab level at 180 days was also still reactive (0.394 µg/mL). We assessed N-Ab levels using an available, fully automated CLIA, which allows for the safer, faster analysis of N-Abs. Our N-Ab findings are supported by another study [17] that assessed 90 BNT162b2 vaccine recipients with another N-Ab CLIA (iFlash-2019-nCoV Nabs assay); N-Ab titers remained reactive (mean 527 AU/mL in initially seronegative individuals) up to 35 days after the second dose of vaccine.

IgM is supposed to define the acute physiological immune response, developing prior to IgG. However, our findings show that even 10 days after the first vaccine dose, a much smaller percentage of subjects were IgM positive compared to IgG (14.1% vs. 69.0%). Even at the peak 20 days after the second dose, only 79.1% were IgM positive. Thus, as a marker for acute immunity, IgM does not appear to be as reactive as IgG post-vaccination. This is also supported in other studies following the development of antibodies post-vaccination [18], where at 1 month post second vaccine dose, BNT162b2 induced a much more robust IgG response than IgM. Even in natural COVID-19 infection, it has been demonstrated that there is variability in the seroconversion of IgG and IgM—occurring synchronously, IgM prior to IgG, or IgG prior to IgM [19].

We also observed that T-Ab and N-Ab levels were still significantly higher 180 days after the second dose of vaccine than 20–40 days post-first dose. The longevity of IgG responses after two doses of vaccine is supported by a British study [20] of 45,965 adults, where even in subjects without prior COVID-19, IgG S-Ab levels after two doses of vaccine remained high even at 91 days from the first vaccination. The greater antibody response after the second dose of vaccine has also been observed in other studies, where subjects without prior COVID-19 required two doses of vaccine to achieve >95% neutralization [21]. This would tend to argue against the use of single-dose vaccine regimens in patients without prior COVID-19, as seronegative subjects mount a significantly lower vaccination response compared to patients with prior COVID-19 [22]. Indeed, some studies [23] show that the effectiveness in preventing hospitalization is only 33% after one dose of BNT162b2 vaccine, and 73% effective only after the second dose. Furthermore, two doses of vaccine seem to be able to neutralize B.1.1.7 variants, with some protection against the B.1.351 variant [24]. Although the median IgG S-Ab was numerically higher at 180 days post-dose 2 than at 20–40 days post-first dose, this did not achieve statistical significance. It is noteworthy that in another study [25] 4 months after the second vaccine dose, IgG S-Ab was lower than the levels 3 weeks after the first vaccine dose.

In our analysis of half-lives of antibodies in our study, we observed that the durations were shorter than those reported in studies using patients with previous COVID-19. In a study [26] of 118 subjects with previous COVID-19, IgG S-Ab (also Abbott) half-lives were estimated to be 198.8 days based on linear regression modelling, nearly double that of IgG Nuc-Ab responses (76.4 days). Our half-lives for antibodies may differ because we used initially seronegative subjects, who may have a different antibody profile to post-COVID-19 infection. In another study that generated a decay model of neutralization titers with time [16], half-lives post mRNA-based vaccination in COVID-naïve subjects was estimated to be 65 days, which was fairly similar to the half-life of antibodies developed post-infection (58 days, *p* = 0.88). The half-lives they generated were more similar to the half-lives we found in our study. Furthermore, the decline is non-linear, with levels plateauing over time. Based on our non-linear regression equation, the N-Ab would decrease to 0.3 µg/mL (reactive cut-off) at 241 days. IgG would decrease to a level corresponding to a positive N-Ab (0.3 µg/mL) by 163 days, with the T-Ab decrease to this corresponding level by 470 days. Even though IgG may decrease earlier, it is reassuring that N-Ab (responsible for most of the immunity) and T-Ab titers remain reactive for a longer period. This provides an estimated time frame for the supposed effectiveness of vaccination in initially seronegative individuals. However, one caveat with basing immune protection on the acute antibody trend is that this does not take into account other immune mechanisms, especially the action of memory B cells, in providing immunity from infection. In post-COVID-19 cases, RBD-specific memory B cell counts remain unchanged even at 6.2 months after infection, with even greater viral neutralization potency [27]. This is also supported by other studies where spike-specific IgG memory B cells increased over time, 4–8 months after COVID-19 infection [28,29]. mRNA vaccines have been shown to produce a similar rise in memory B cells in seronegative individuals after two doses of vaccine [30], and further studies are required to prove the efficacy of these memory B cell responses.

An additional finding in our study is that although we previously observed a significant difference in antibody titers between subjects <50/≥50 years old prior to 90 days post-vaccination [3], we did not find any significant difference in the antibody titers between age groups after 90 days. This finding persisted even when we evaluated results from each time point in isolation (120/160/180). Other studies have shown that antibody levels decrease with age, even in previously uninfected individuals [22]. Another study [31] that evaluated the vaccine responses between those <60 and >80 years old found that younger subjects had significant higher IgG S-Ab (Euroimmun) and neutralization titers (in house neutralization assay) 17 days after the second mRNA vaccine dose. One possible reason why we were unable to find a difference may be that the two groups in our study may be too close in age and that we did not have sufficient numbers of older subjects.

One limitation of our study is that we have few patient samples 20–40 days after the first vaccination (*n* = 8), as well as at 180 days post-vaccination (*n* = 8), as not all vaccinees had an extended period between vaccinations, and more participants are still waiting to reach the 180-day post-vaccination mark. The 8 subjects at these points were not the same individuals. The small number of subjects at these time points may affect the antibody levels at these two time points, and further studies with larger numbers of subjects would be desirable. Our study population had a preponderance of women (75.3%, 58/77) and younger subjects <50 years old (72.7%, 56/77). We were also unable to obtain samples from vaccinated individuals who had previous COVID-19, nor were we able to study changes of antibody titers against other coronaviruses.

## 5. Conclusions

We have demonstrated that even in COVID-naïve individuals, T-Ab, IgG and N-Ab titers remain high after two doses of mRNA vaccine, even up to 180 days, with T-Ab and N-Ab levels still higher at 180 days than 20–40 days after a single dose of vaccine. IgM does not necessarily develop early post-vaccination, with more subjects having a positive IgG at each time point. Non-linear regression models show that after the second vaccination, N-Ab responses should remain reactive to around 241 days, suggesting a prolonged duration of immunity.

## Figures and Tables

**Figure 1 vaccines-09-01241-f001:**
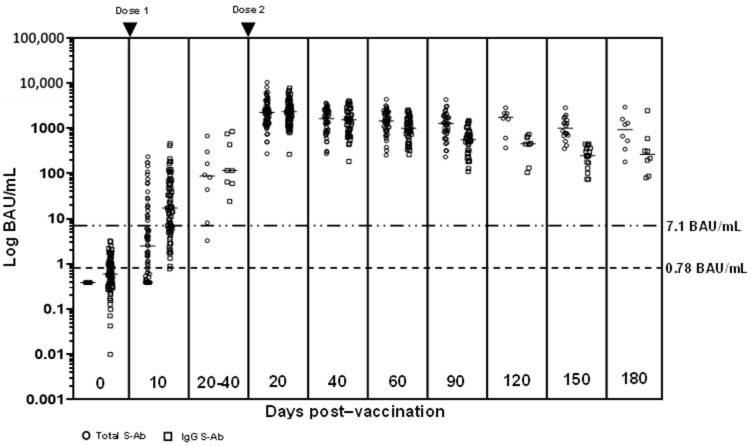
Total (Roche Cobas) and IgG (Abbott Architect) spike antibodies from baseline to 180 days after complete vaccination. Total/IgG spike antibodies are positive at ≥0.78/≥7.1 BAU/mL.

**Figure 2 vaccines-09-01241-f002:**
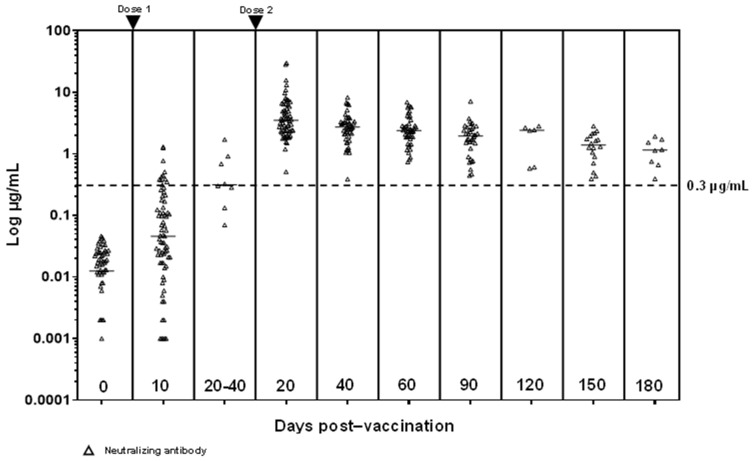
Antibody progression of neutralizing antibodies (Snibe Maglumi) from baseline to 180 days after the second vaccine dose. Neutralizing antibodies are positive at ≥0.3 µg/mL.

**Figure 3 vaccines-09-01241-f003:**
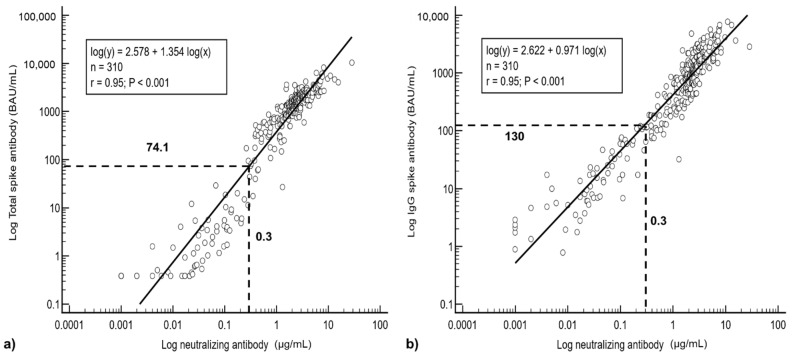
Linear regression analysis between (**a**) total and (**b**) IgG spike antibodies with neutralizing antibodies. Based on linear regression analysis, a total spike antibody level of 74.1 BAU/mL and an IgG spike antibody level of 130 BAU/mL would correspond to the reactive limit of the neutralizing antibody assay (≥0.3 µg/mL).

**Figure 4 vaccines-09-01241-f004:**
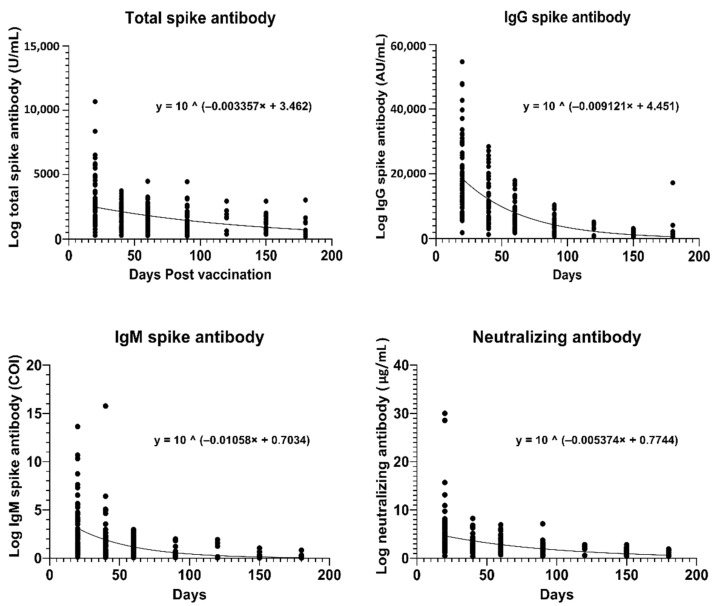
Non-linear regression models for total, IgG, IgM spike antibodies and neutralizing antibodies after the second vaccination dose.

**Table 1 vaccines-09-01241-t001:** Estimated antibody levels post-vaccination based on the non-linear regression models, with estimated half-lives and days to (corresponding) N-Ab cut-off.

Days Post-Vaccination	20	40	60	90	120	150	180	Half-Life	Days to N-Ab Cut-Off *
Total (U/mL)	2482	2127	1822	1445	1146	909	721	90 days	470
IgG (AU/mL)	18,560	12,194	8012	4267	2272	1210	644	33 days	163
N-Ab (µg/mL)	4.64	3.63	2.83	1.95	1.35	0.93	0.64	56 days	241

* Days to N-Ab cut-off: Based on linear regression models, a total spike antibody level of 76.4 U/mL (74.1 BAU/mL) and an IgG spike antibody level of 916 AU/mL (130 BAU/mL) would correspond to the neutralizing antibody reactive cut-off of ≥0.3 µg/mL. These values were substituted into the equations of our non-linear regression model to determine the number of days post-vaccination total and IgG spike antibodies would require to reach the levels that corresponded to the neutralizing antibody cut-off.

## Data Availability

The datasets generated during and/or analyzed during the current study are not publicly available due to privacy issues and national laws but are available from the corresponding author on reasonable request under the provision that data may not leave the hospital/center premises.

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
