# Peer review of "Robust SARS-CoV-2 Antibody Responses in Asian COVID-Naïve Subjects 180 Days after Two Doses of BNT162b2 mRNA COVID-19 Vaccine"

_vaccines, 2021, doi:10.3390/vaccines9111241_

Round 1

Reviewer 1 Report

The authors present results of antibody responses to mRNA COVID-19 vaccine 180 days after the second dose in persons of Asian origin. The study is well designed and executed. In addition, it addresses an important issue of humoral immunity induced by mRNA vaccine in a group of people with defined ethnic background. I recommend the publication of this study in the present form. 

Reviewer 2 Report

In this paper, the authors have analyzed antibody response in 77 healthcare workers , who were naïve of SARS-CoV-2 infection, and vaccinated by the BNT162b2 mRNA COVID-19 vaccine.

Major comments:

  • The authors explain they performed this study on subjects from Southern Asia since it has not be studied hitherto. The authors should explain why they think that Asian people have a different serological response to COVID vaccination that caucasian/american people. In addition, their results should be compared in the discussion section to the serological response in white people.
  • The authors explain that because different vaccination schedules, the time at each time point is different. However, they should explain why they have only 8 volunteers at D180. This an important limitation to their conclusions. When they compare antibody titers between D20-40 and D180, there is only 8 patients in each group...It should be indicated if it is the same groupe of volunteers and that is a limitation to their study.
  • Discussion is too long. Page 8, line 290, there is a discussion on the interest to use 2 doses of vaccin, but it is largely admitted by the WHO and the different governmental authorities all over the world that 2 doses are needed with the BTN162b2 vaccin. Consequently this discussion is not relevant. The paragraph from line 337 to line 351 is a repetition of the results, and largely redundant with the conclusion (paragraph 5.0)

Minor comments

  • the reagents used fo the determination total S-Ab, Ig S-Ab and N-Ab should be indicated in the legends of the figures 1 & 2
  • Abbreviations section: a l is missing for confidence interval.

Author Response

Comment 1: The authors should explain why they think that Asian people have a different serological response to COVID vaccination that causcasian/American people. In addition, their results should be compared in the discussion section to the serological response in white people.

Reply: We have added additional discussion on this point in line 260.

Comment 2: However, they should explain why they have only 8 volunteers at D180. This is an important limitation to their conclusions. When they compare antibody titers between D20-40 and D180, there is only 8 patients in each group. It should be indicated if it is the same group of volunteers and that is a limitation to their study

Reply: We have elaborated and acknowledged this limitation in line 349.

Comment 3: There is a discussion on the interest to use 2 doses of vaccine, but it is largely admitted by the WHO and the different governmental authorities all over the world that 2 doses are needed with the BNT162b2 vaccine.

Reply: Although it is widely accepted that 2 doses are needed for the Pfizer vaccine, there are still many developing nations that are struggling to administer 2 doses to their populace. It is still important to reinforce the futility of only using one dose of vaccine. As such, we still include the discussion from line 295-311.

Comment 4: The paragraph from line 337 to line 351 is a repetition of the results, and largely redundant with the conclusion.

Reply: We have acknowledged this suggestion, and have removed that section from the discussion

Comment 5: The reagents used for the determination of total S-Ab, IgG S-Ab and N-Ab should be indicated in the legends of the figures 1 & 2.

Reply: We have included the assays in the figure legends.

Reviewer 3 Report

This is an interesting manuscript which however needs more data.

Introduction:it should nbe made clear that we do not really know how the vaccine works.In fact,nobody has ever shown the protein coded by the mRNA contained in the lipid particles.The particle are most probably taken up by the macrophages of the regional lymphnodes.It is also not clear how the measured antibody protect from the viral infection as the virus doies not reach the circulation.The use of the definition of "neutralizing" antibody may be meslaeding as this is the result of an in vitro-test.

Possible changes of antibody titer (not only against the spike protein) against other coronaviruses against e.g O43 should also be studied

Author Response

Comment 1: It should be made clear that we do not really know how the vaccine works. In fact, nobody has ever shown the protein coded by the mRNA contained in the lipid particles.

Reply: We have now acknowledged this in line 55.

Comment 2: The use of the definition of “neutralizing” antibody may be misleading as this is the result of an in vitro-test.

Reply: We have elaborated upon this in line 64-71.

Comment 3: Possible changes of antibody titer (not only against the spike protein) against other coronaviruses against e.g. O43 should also be studied.

Reply: We have acknowledged this limitation on line 357.

Round 2

Reviewer 3 Report

The changes introduced increased the quality of this manuscript which is now

cceptable for publication